# "*Separated during the first hours*"—Postnatal care for women and newborns during the COVID-19 pandemic: A mixed-methods cross-sectional study from a global online survey of maternal and newborn healthcare providers

**Aline Semaan**[1]*, **Teesta Dey**[2], **Amani Kikula**[3], **Anteneh Asefa**[1], **Thérèse Delvaux**[1], **Etienne V. Langlois**[4], **Thomas van den Akker**[5,6], **Lenka Benova**[1]

1 Department of Public Health, Institute of Tropical Medicine, Antwerp, Belgium, 2 Department of Women's and Children's Health, University of Liverpool, Liverpool, United Kingdom, 3 Muhimbili University of Health and Allied Sciences, Upanga-West, Dar es salaam, Tanzania, 4 Partnership for Maternal, Newborn and Child Health (PMNCH), World Health Organization (WHO), Geneva, Switzerland, 5 Faculty of Science, Athena Institute, Vrije Universiteit Amsterdam, Amsterdam, The Netherlands, 6 Department of Obstetrics and Gynaecology, Leiden University Medical Centre, Leiden, The Netherlands

* asemaan@itg.be

## Abstract

Routine postnatal care (PNC) allows monitoring, early detection and management of complications, and counselling to ensure immediate and long-term wellbeing of mothers and newborns; yet effective coverage is sub-optimal globally. The COVID-19 pandemic disrupted availability and quality of maternal and newborn care despite established guidelines promoting continuity of essential services. We conducted a cross-sectional global online survey of 424 maternal and newborn healthcare providers from 61 countries, to explore PNC provision, availability, content and quality following the early phase of the COVID-19 pandemic. The questionnaire (11 languages), included four multiple-choice and four open-text questions on changes to PNC during the pandemic. Quantitative and qualitative responses received between July and December 2020 were analysed separately and integrated during reporting. Tightened rules for visiting postpartum women were reported in health facilities, ranging from shorter visiting hours to banning supportive companions and visitors. A quarter (26%) of respondents reported that mothers suspected/confirmed with COVID-19 were routinely separated from their newborns. Early initiation of breastfeeding was delayed due to waiting for maternal SARS-CoV-2 test results. Reduced provision of breastfeeding support was reported by 40% of respondents in high-income countries and 7% in low-income countries. Almost 60% reported that women were discharged earlier than usual and 27% perceived a reduction in attendance to outpatient PNC. Telemedicine and home visits were mostly reported in high-income countries to ensure safe care provision. Beyond the early phase of the COVID-19 pandemic, severe disruptions to content and quality of PNC continued to exist, whereas disruptions in availability and use were less commonly reported. Depriving women of support, reducing availability of PNC services, and mother-newborn separation could lead to negative long-term outcomes for women,

**Data Availability Statement:** Due to ethical constraints, the data underlying this analysis cannot be made publicly available. The dataset cannot be completely de-identified without removing key variables such as country, cadre, facility level, facility sector, area type and all the open-ended questions. This de-identification would limit the value of the dataset, making any replication of the analysis impossible. Data requests can be sent to the study PI Prof Lenka Benova at lbenova@itg.be and the ethics committee at the Institute of Tropical Medicine at irb@itg.be.

**Funding:** This study was funded by the Institute of Tropical Medicine's COVID-19 Pump Priming fund supported by the Flemish Government, Science & Innovation and by the Embassy of the United Kingdom in Belgium. LB is funded in part by the Research Foundation – Flanders (FWO) as part of her Senior Postdoctoral Fellowship. The funders had no role in study design, data collection and analysis, decision to publish, or preparation of the manuscript.

**Competing interests:** The authors have declared that no competing interests exist.

newborns and families, and deny their rights to respectful care. Protecting these essential services is imperative to promoting quality woman-centred PNC during and beyond the pandemic.

## Introduction

The postnatal period, defined as the time immediately following childbirth until 42 days after the birth, is a critical time when substantial support and care are needed for both women and newborns [1]. The largest burden of mortality and morbidity for mothers and babies occurs within the postnatal period, especially in the immediate period after birth [2,3]. Globally, this burden is unevenly distributed, whereby 90% of maternal deaths in the postpartum period are estimated to occur in Sub-Saharan Africa and Southern Asia [4]. In this paper, the term "post-partum period" refers to issues pertaining to the mother and the term "postnatal period" refers to those concerning the baby. For clarity, the term "postnatal care" is used to refer to the care after childbirth for both the mother and newborn.

Currently, the World Health Organization (WHO) recommends a minimum hospital stay of 24 hours after uncomplicated vaginal birth in a health facility. In case of home births, a first postnatal contact with a skilled provider should occur within the first 24 hours, either at home or in a health facility. In both cases, a minimum of three postnatal checks for both mother and newborn should follow: within the first three days after birth, in the second week after birth, and six weeks after birth [1]. These consecutive visits offer opportunities for routine monitoring, early detection, recognition and treatment of conditions (e.g. postpartum haemorrhage, infection, venous thromboembolism for the mother, and sepsis and jaundice for the baby) which can occur at different time points in the postnatal period, and therefore decrease the risk of maternal and neonatal morbidity and mortality. The benefits of routine visits can include providing counselling on breastfeeding, family planning, maternal nutrition, and psycho-social wellbeing of the mother, including exposure to domestic abuse [1]. Additionally, routine postnatal care offers an opportunity to integrate mental health screening, prevention and treatment which can promote the wellbeing of the mother, improve mother-child interactions, and reduce the occurrence of postpartum depression [5–9]. Despite its importance, globally, PNC service coverage is among the lowest on the continuum of maternal and child healthcare [10–12].

Since March 2020, the COVID-19 pandemic has had considerable direct and indirect impacts on the health of pregnant women, mothers and newborns. A living meta-analysis shows that contracting COVID-19 during pregnancy increases the odds of maternal death and preterm birth [13]. The indirect impact of the pandemic is beyond the impact of COVID-19 during pregnancy, since it affects all mothers and newborns, and not just those infected with the virus. Stringent lockdown measures, curfews, and restrictions on transportation have reduced availability, utilisation and quality of essential maternal and newborn health services, including PNC [14–18]. A recent systematic review showed an overall increase in stillbirths and maternal deaths, and considerably worsened maternal mental health during the COVID-19 pandemic [19]. These negative outcomes in many cases resulted from preventable causes or were due to delays in care provision during lockdowns [20,21].

During the first quarter of 2021, 33% of 110 countries reported disruptions to PNC services for women and newborns, and 2% reported more than a 50% disruption [22]. Staff shortages affected the availability of PNC, with newborn care staff being reassigned to COVID-19 related

care [17,18]. Clinical management of small and sick babies in newborn intensive care units (NICUs) and the implementation of Kangaroo Mother Care (KMC) were delayed as a result of the long waiting time to obtain results of SARS-CoV-2 tests [17]. Early discharge from facilities after birth and reduced face-to-face routine visits, and introduction of virtual care (telemedicine) in the postnatal period were encouraged to decrease the risk of infection transmission. Potentially, these adaptations could contribute to missing danger signs and further worsening maternal and newborn health outcomes, including maternal mental health [23,24]. Women and newborns spent less time in the health facility and were encouraged to leave early [16–18,25], early initiation of breastfeeding was delayed, skin-to-skin contact reduced, and KMC practice suspended [17,21].

Some of these disruptive practices specifically targeted women infected with SARS-CoV-2 and their babies [17,18]. Rollins et al. estimated that the separation of mothers confirmed with COVID-19 from their newborns and not allowing breastfeeding would lead to 100 times more infant deaths than the number of infant deaths directly due to COVID-19 in low- and middle-income countries [26]. Evidence exists that the benefits of sustained PNC, including breastfeeding and KMC for preterm babies, far outweigh the risks of COVID-19 infection [27,28]. Therefore, in as early as June 2020, the WHO considered the complete package of PNC as an essential service during the pandemic, and issued practical guidance on how to ensure the continuity of breastfeeding and non-separation for all mothers and newborns [29,30]. Yet, many national-level guidelines still contained recommendations that are against practices supportive of breastfeeding [31], and healthcare providers struggled with the ambiguity of the guidance they were receiving, which impacted care provision [16,18,28].

Evidence suggests considerable collateral damage from the pandemic on PNC coverage and quality and subsequent impact on maternal and newborn outcomes. However, most of these data are strictly quantitative in nature and were collected in the early phase of the pandemic (between March and June 2020). This study's objective was to fill this knowledge gap using data from a mixed methods global online survey of maternal and newborn healthcare providers, conducted after the early phase of the pandemic (between July and December 2020). We present findings describing how mitigation measures implemented to control the spread of COVID-19, (such as strict lockdowns, travel bans and interruption to public transport systems, curfews and closure of health facilities in certain countries [32]) affected the provision, content and quality of PNC. Results were stratified by country income groups to help us understand the findings contextually considering available resources and capacities.

## Methods

### Study design

This study uses data from a global online survey of maternal and newborn healthcare providers during the COVID-19 pandemic. It was administered as three repeated cross-sectional surveys at different time points. In this paper, we present findings from submissions made to the second round of the survey, between July 5, 2020 and December 14, 2020. The survey targeted maternal and newborn healthcare providers from various cadres, including midwives, nurses, obstetricians/gynaecologists, neonatologists and paediatricians, among others. We recruited participants using two approaches: 1) we sent email invitations to participants who answered the first round of the survey and who agreed to be contacted for upcoming rounds; 2) we widely disseminated the survey to other maternal and newborn healthcare providers by distributing the link to the online questionnaire through personal and professional networks, and social media channels (e.g. Twitter, Facebook, WhatsApp groups, etc.). Additional details about the study design and sampling are available elsewhere [18].

## Questionnaire

The questionnaire of the second survey round was developed by adapting some questions from the first round questionnaire based on the responses that we received. The multidisciplinary study team who contributed to the survey development and amendments included health professionals and experts in health systems, maternal and newborn health epidemiologists and public health researchers, acknowledged previously [33].

We maintained the core themes of the questionnaire by asking respondents about their background, preparedness for and response to COVID-19, and their own work experience in the month preceding the time they answered the survey. Participants were given the opportunity to opt in or out of answering an optional module of 11 questions about their perceptions of how the COVID-19 pandemic had altered facility- and community-based care provided to women and newborns during the month preceding the time they answered the survey. This section included questions about services along the continuum of maternal and newborn care (family planning, antenatal, intrapartum, postnatal, abortion and post-abortion care), provided both in inpatient and outpatient settings, and specifically about changes to the care provision process, targeting the content and quality of care.

For the purpose of this paper, we use answers to the four multiple choice questions related to PNC: 1) inpatient PNC process; 2) content and quality of inpatient PNC; 3) outpatient PNC process; 4) content and quality of outpatient PNC for women and newborns. The distinction between inpatient and outpatient services does not necessarily reflect the timing of the postnatal check (immediate vs. long-term) considering the variability in postnatal length-of-stay between countries and facilities. Each question had a set of multiple choice options which were developed based on the themes that emerged from the first round [18]. Participants had the option to select multiple answers to each question, and therefore the answers were not mutually exclusive. Additionally, following each question there was an open text field for respondents to describe in detail and elaborate on the changes that occurred as a result of the pandemic. Aspects related to quality of care were mostly derived from open-text responses. The full questionnaire is available on the study website [34], and the questions relevant to this analysis are provided in S1 Table.

The questionnaire was published online using KoboToolbox's online data collection feature [35], and it was available in 11 languages (Arabic, Dutch, English, French, German, Italian, Japanese, Kiswahili, Portuguese, Russian and Spanish).

## Data management and analysis

A total of 1,405 survey responses were received through the online platform during the survey period. Almost a third of these respondents agreed to participate in the optional module (n = 443, 31.5%). In this analysis, we included data from 424 (95.7% of the optional module respondents) healthcare providers who answered at least one of the four questions on PNC (19 respondents were removed from this analysis because they did not answer at least one of the four questions on PNC). The country income level variable (high income, middle income, low income) was added to the dataset post-hoc using the World Bank classification of the worlds' economies (according to 2020 gross national income) [36]. Quantitative and qualitative data collection and analysis were performed simultaneously in a convergent mixed-methods design [37]. Quantitative analysis involved producing descriptive statistics (frequencies and percentages), disaggregated by country income level, using Stata/SE version 16. The number of missing values varied between questions, and therefore so did the denominator; we report the final sample size for each variable after removing missing data. We analysed open-ended answers using qualitative content analysis. A preliminary list of codes was developed based on the

themes in the questionnaire as per the objectives of the study (deductive), and expanded with new codes and themes that came up in the data (inductive). A final list of codes was reached and a coding framework describing the changes and adaptations made to PNC was developed by consensus following a discussion among the study team. One or multiple codes were applied to each open-ended answer. If a code corresponded to one of the concepts listed in the close-ended question options, an additional observation was added to the final count of each option. Findings from quantitative and qualitative data were summarised as themes, and were interpreted and reported in an integrated manner, with quotes used to illustrate examples on the quantitative data. Data were coded by one researcher who is fluent in English, Arabic and French (AS), and the codes and themes were discussed with the research team.

## Ethics

This study was approved by the Institutional Review Board at the Institute of Tropical Medicine in Antwerp Belgium under the number 1372/20. Respondents provided informed consent online by checking a box affirming that they voluntarily agreed to participate in the survey.

## Results

### Sample characteristics

The 424 respondents worked in 61 countries; almost half (n = 193/424, 46%) of them were from high-income countries, 42% (n = 181/424) from middle-income countries and the minority worked in low-income countries (n = 50/424, 12%, Table 1), of which half provided care in the Democratic Republic of the Congo (S2 Table). Midwives and obstetricians/gynae-cologists each comprised about a quarter of the sample (n = 113/418, 27% for each cadre). The largest group of respondents from high-income countries were midwives (n = 92/190, 48%), whereas in low-income countries, 55% (n = 27/49) were medical doctors (Table 1). One in ten of respondents provided only PNC (n = 43/404, 10%), whereas almost a quarter did not provide any healthcare services in the postnatal period, including newborn care (n = 106/404, 26%). The majority of respondents provided PNC in combination with other maternal or newborn health services (e.g. antenatal care, intrapartum care, family planning counselling, etc.) in a health facility setting (n = 155/404, 38%) and/or in a home/community setting (n = 100/404, 25%; Table 1). Most of the respondents provided care in hospitals (n = 219/415, 53%), and around 13% (n = 50/415) were independent or self-practicing, while less than 1% mentioned that they provide home-based care (n = 3/415). The majority of healthcare providers worked in public facilities (n = 301/415, 73%) and urban settings (n = 338/411, 82%).

The findings from the optional module regarding changes to the provision of PNC to women and newborns are summarised in three main themes: 1) adaptations in the process of care aiming to reduce the risk of virus transmission and infection; 2) changes in PNC use and provision and service availability; and 3) modifications to the content and quality of care provided to all women and newborns, and specifically to women and newborns suspected/confirmed to have COVID-19.

1- **Adaptations related to infection prevention and control measures.** The most commonly reported changes to the immediate postpartum period related to visiting rules in health facilities, including a reduction in the number of visitors allowed, banning visitors altogether, and shortening visiting hours (reported by n = 219/397 (55%), n = 168/397 (42%) and n = 150/397 (38%) respectively; Fig 1). Differences were noted between countries by income level, with complete banning of visitors reported by two thirds (n = 115/180; 64%) of the respondents in high-income countries compared to 6% (n = 3/49) of respondents in

**Table 1.  Characteristics of the sample of maternal and newborn healthcare providers (n = 424)\*, column percentages.**

| | HICs (n = 193; 45.5%) | MICs (n = 181; 42.7%) | LICs (n = 50; 11.8%) | Total (n = 424; 100%) |
|---|---|---|---|---|
| **Cadre** | **(n = 190)** | **(n = 179)** | **(n = 49)** | **(n = 418)** |
| Midwife | 92 (48.2) | 18 (10.0) | 3 (6.1) | 113 (27.0) |
| Obstetrician/gynaecologist | 41 (21.6) | 64 (35.8) | 8 (16.3) | 113 (27.0) |
| Medical doctor | 3 (1.6) | 29 (16.2) | 27 (55.1) | 59 (14.1) |
| Nurse | 13 (6.8) | 36 (20.1) | 6 (12.2) | 55 (13.2) |
| Nurse-midwife | 20 (10.5) | 12 (6.7) | 1 (2.0) | 33 (7.9) |
| Neonatologist/Paediatrician | 20 (10.5) | 12 (6.7) | 0 (0) | 32 (7.7) |
| Other | 1 (0.5) | 8 (4.4) | 4 (8.2) | 13 (3.1) |
| **Position** | **(n = 191)** | **(n = 174)** | **(n = 49)** | **(n = 414)** |
| Head of facility | 7 (3.7) | 12 (6.9) | 6 (12.2) | 25 (6.0) |
| Head of department or ward | 18 (9.4) | 35 (20.1) | 15 (30.6) | 68 (16.4) |
| Head of team | 26 (13.6) | 22 (12.6) | 11 (22.5) | 59 (14.3) |
| Team member | 96 (50.3) | 43 (24.7) | 12 (24.5) | 151 (36.5) |
| Independent or self-practicing | 36 (18.9) | 17 (9.8) | 3 (6.1) | 56 (13.5) |
| Locum or interim member | 3 (1.6) | 11 (6.3) | 0 (0) | 14 (3.4) |
| Other | 5 (2.6) | 34 (19.5) | 2 (4.1) | 41 (9.9) |
| **Type of care provided by respondent** | **(n = 192)** | **(n = 163)** | **(n = 49)** | **(n = 404)** |
| Postnatal care only | 16 (8.3) | 21 (12.9) | 6 (12.2) | 43 (10.6) |
| Postnatal care and other services in health facilities | 82 (42.7) | 59 (36.2) | 14 (28.6) | 155 (38.4) |
| Postnatal care and other services in homes/community | 50 (26.0) | 35 (21.5) | 15 (30.6) | 100 (24.8) |
| No postnatal care services | 44 (22.9) | 48 (29.5) | 14 (28.6) | 106 (26.2) |
| **Type of the facility where respondents work** | **(n = 190)** | **(n = 176)** | **(n = 49)** | **(n = 415)** |
| Referral hospital | 50 (26.3) | 51 (28.9) | 20 (40.8) | 121 (29.2) |
| District/regional hospital | 67 (35.3) | 28 (15.9) | 3 (6.1) | 98 (23.6) |
| Polyclinic or clinic | 7 (3.7) | 38 (21.6) | 8 (16.3) | 53 (12.8) |
| Health centre | 15 (7.9) | 24 (13.6) | 3 (6.1) | 42 (10.1) |
| Birth centre | 7 (3.7) | 7 (3.9) | 0 (0) | 14 (3.4) |
| Health post/unit or dispensary | 0 (0) | 4 (2.3) | 0 (0) | 4 (1.0) |
| Home-based care | 2 (1.1) | 1 (0.6) | 0 (0) | 3 (0.7) |
| Independent or self-practicing | 35 (18.4) | 12 (6.8) | 3 (6.1) | 50 (12.1) |
| Other | 7 (3.7) | 11 (6.2) | 12 (24.5) | 30 (7.2) |
| **Facility sector** | **(n = 189)** | **(n = 176)** | **(n = 50)** | **(n = 415)** |
| Public (national) | 61 (32.3) | 87 (49.4) | 18 (36.0) | 166 (40.0) |
| Public (university or teaching) | 29 (15.3) | 32 (18.2) | 11 (22.0) | 72 (17.4) |
| Public (district level or below) | 34 (18.0) | 24 (13.6) | 5 (10.0) | 63 (15.2) |
| Private | 23 (12.7) | 9 (5.1) | 5 (10.0) | 37 (8.9) |
| Non-governmental or faith based organisation | 2 (1.1) | 5 (2.8) | 6 (12.0) | 13 (3.1) |
| Health insurance/Social security | 4 (2.1) | 7 (4.0) | 0 (0) | 11 (2.7) |
| Independent or self-practicing | 34 (18.0) | 7 (4.0) | 2 (4.0) | 43 (10.4) |
| Other | 2 (1.1) | 5 (2.8) | 3 (6.0) | 10 (2.4) |
| **Type of geographic area where respondents work** | **(n = 187)** | **(n = 175)** | **(n = 49)** | **(n = 411)** |
| Large city (> 1 mil inhabitants) | 48 (25.8) | 82 (46.9) | 24 (49.0) | 154 (37.5) |
| Small city (100,000 to 1 mil inhabitants) | 69 (36.9) | 34 (19.4) | 12 (24.5) | 115 (28.0) |
| Town (<100,000 inhabitants) | 49 (26.2) | 18 (10.3) | 2 (4.1) | 69 (16.8) |
| Village or rural area | 19 (10.2) | 35 (20.0) | 10 (20.4) | 64 (15.8) |
| Other | 2 (1.1) | 6 (3.4) | 1 (2.0) | 9 (2.2) |

*(Continued)*

**Table 1.** (Continued)

| | HICs (n = 193; 45.5%) | MICs (n = 181; 42.7%) | LICs (n = 50; 11.8%) | Total (n = 424; 100%) |
|---|---|---|---|---|
| Total countries | 27 HICs | 23 MICs | 11 LICs | 61 countries |

*Varying number of missing values in each variable.

low-income countries. Reducing the number of allowed visitors was more commonly reported by respondent in low-income countries (n = 35/49; 71%), compared to 59% (n = 107/180) and 46% (n = 77/168) of respondents from high-income and middle-income countries respectively. In the open-text responses, healthcare providers elaborated that these rules were in some cases applied with exceptions as to the type of visitor. Some respondents noted that the visiting ban only applied to family members and friends, and excluded partners (e.g. spouse). Other respondents mentioned that allowing companions to stay overnight was possible with certain logistical constraints. A midwife in Iceland explained that "*partners [are] not allowed to stay overnight in the postnatal unit unless the woman had a single room*". Rules on visitors in the postnatal period were in many cases an extension to the rules on companionship during labour and childbirth (See Box 1). A midwife in Argentina listed a number of conditions that made it difficult for birth companions to stay in the postnatal ward, including the requirement to present a negative COVID-19 test: "*The father can only enter at birth if he is negative [for SARS-CoV-2] and stays throughout the hospitalization without leaving the room and brings his own food. This means that no one can stay... It is a trap to abuse rights and cover up the prohibition of companionship.*" One respondent from the UK reported that the absence of visitors and companions had a positive impact on breastfeeding as there were fewer distractions, and staff had more time to dedicate to women. Almost 12% (n = 49/397) of the respondents reported that parents

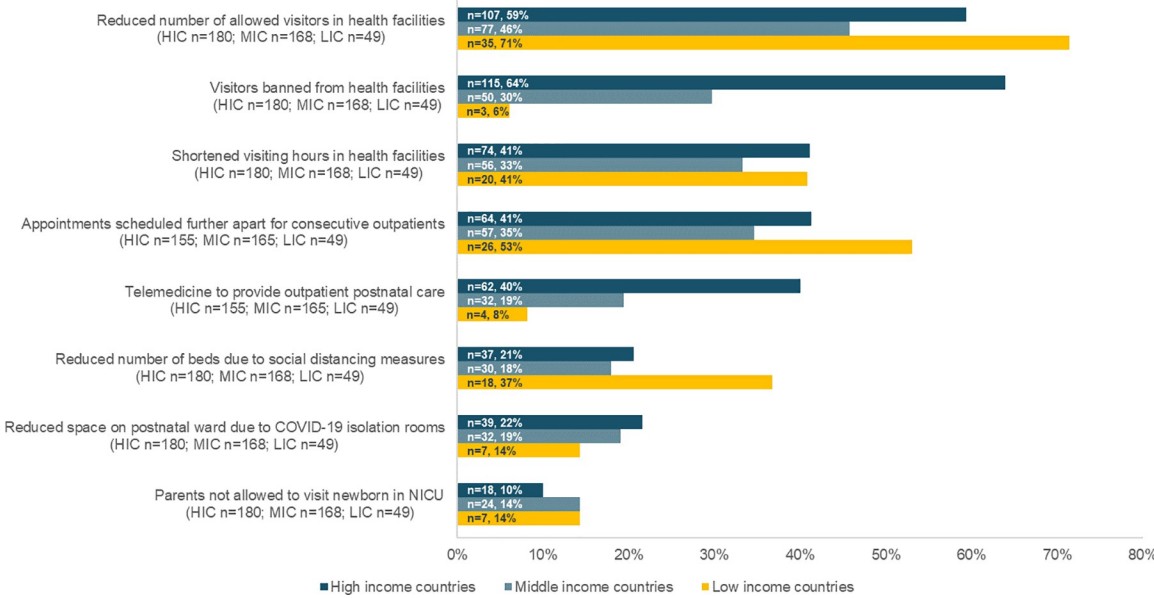

**Fig 1. Number and percentage of respondents reporting adaptations in the process of postnatal care aiming to reduce the risk of SARS-CoV-2 transmission, by country income level during the month preceding the survey.**

were banned from visiting their babies in NICU (Fig 1). On the contrary, few respondents to the open-text questions noted that the month preceding the survey involved a relaxation of the previously implemented visitor restrictions in postnatal wards. A nurse-midwife from the USA expressed: "*Visitor restrictions [are] lifted somewhat, now allowed two visitors vs. previously allowed one*".

---

### Box 1. Linkages between limitations on birth companionship and postnatal visiting between July and December 2020

Among the series of questions in the optional module, healthcare providers were asked about changes to the rules on birth companionship for all women during the month preceding the survey. The World Health Organization's intrapartum care guidelines recommend that women be accompanied by birth companions of choice during labour and childbirth, including for women suspected or confirmed with COVID-19 [38–40]. In this study, 90/346 respondents (26%) reported that companions were banned for all women during this period.

Fig 2 shows that among healthcare providers who reported a ban on birth companionship, the majority (n = 54/90, 60%) also noted that visitors were not allowed in the postnatal ward. In answers to open-ended questions, some respondents elaborated that the ban on birth companionship was implemented with a restriction on the identity of the visitors to the postnatal ward, permitting only partners or spouses.

A nurse-midwife in Japan wrote: "*As a general rule, visitors are limited to husbands, and visits to children's grandparents and siblings are prohibited*". Even among the respondents who reported that there was no ban on birth companionship, 35% (n = 91/256) reported that there was a ban on visitors (Fig 2). In these cases, the partners were mainly allowed to accompany the women during childbirth and other visitors (family, friends, etc.) were banned.

---

Fig 1 also shows that the availability of space and beds for immediate PNC provision was also affected during the study period. Overall, around one-fifth of respondents reported that the adopted infection prevention measures such as social distancing meant that there were fewer beds in their facility's postnatal ward; this was reported by 37% (n = 18/49) of respondents in low-income countries. The creation of COVID-19 isolation rooms contributed to reducing the space available in postnatal wards, as reported by n = 39/180 and 32/168 of respondents in high-income and middle-income countries respectively, compared to 14% (n = 7/49) of respondents in low-income countries. Common adaptations to long-term postnatal care included scheduling consecutive outpatient PNC appointments for different women further apart in time, which was reported by around two fifths of the respondents (n = 147/369). The provision of PNC through telemedicine was reported by 40% (n = 62/155) of the respondents in high-income countries and 8.2% (n = 4/49) of the respondents in low-income countries. Self-monitoring was also practiced to a certain extent, as a midwife in Canada mentioned that she "*taught parents to monitor infant weight*".

**2- Postnatal care use and provision and service availability.** Changes to PNC availability and use were not commonly reported and mainly included the prioritisation of providing care to patients with the highest needs (n = 121/369; 33%). Moreover, 27% (n = 99/369) of the

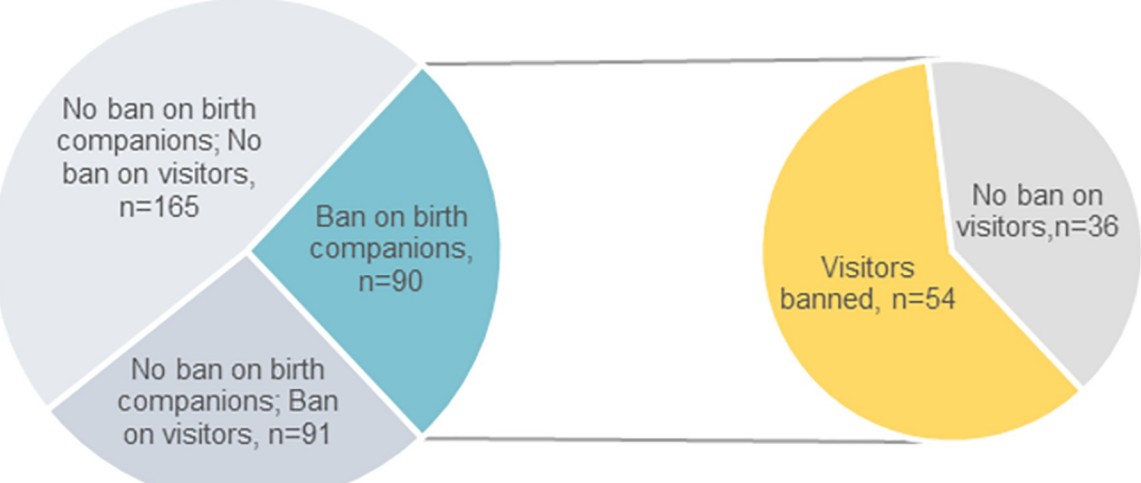

**Fig 2. Breakdown of the number of respondents who mentioned that there was a ban on companionship during childbirth by their response regarding the ban on visitors in the postnatal period (n = 346).**

respondents perceived a reduction in the number of women and newborns accessing outpatient PNC services. These two changes were mentioned by a higher proportion (n = 21/49; 43% and n = 18/49; 37%, respectively) of respondents from low-income countries compared to respondents from middle- and high- income countries (Fig 3).

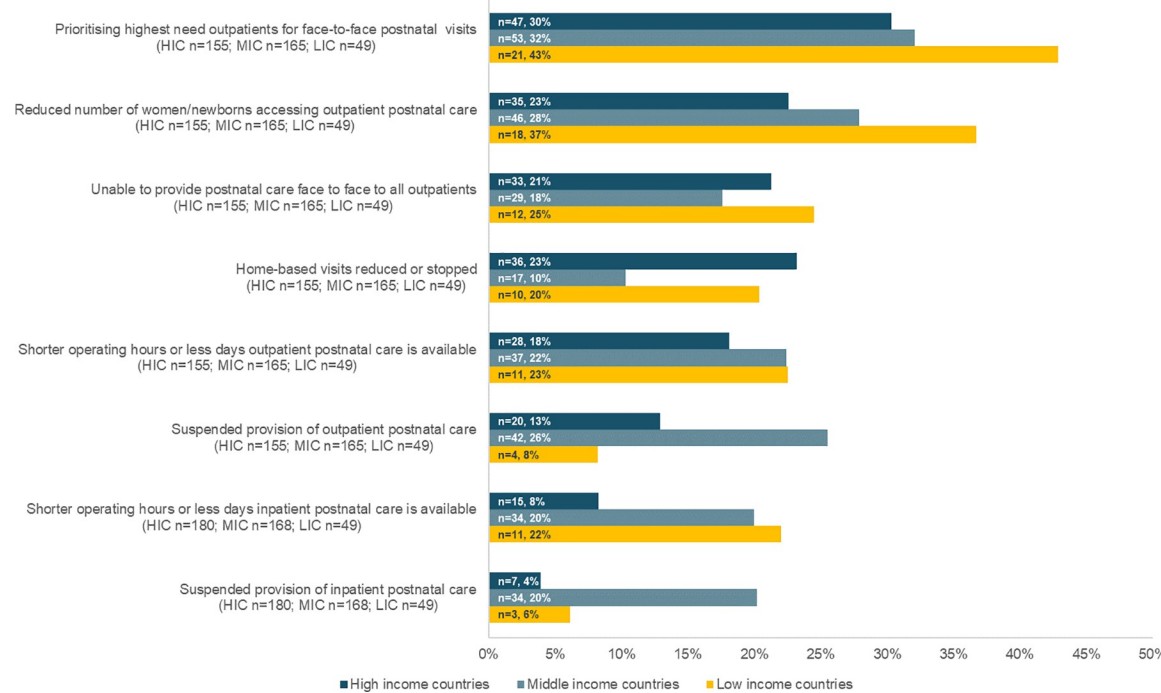

**Fig 3. Number and percentage of respondents reporting changes in postnatal care use and provision and service availability, by country income level during the month preceding the survey.**

According to 17% (n = 63/369) of respondents, home-based PNC visits were reportedly reduced or stopped during the pandemic (Fig 3). On the other hand, some responses to the open-text show that home visits in the postnatal period were increased during the study period as they helped in reducing the crowding in the waiting areas of the outpatient clinics. A midwife who works in Belgium wrote: "*We added the three-week home visit to empty the clinic waiting room because these patients would have had to visit the clinic at three weeks postpartum before the pandemic*". Additionally, home visits were provided to compensate for the needed care in case women were discharged early, and to supplement the care provided by telemedicine, as a midwife in Germany stated: "*In the case of a video appointment, I made a home visit in the following days if problems couldn't be solved otherwise.*" This modality of care provision was not without some challenges to healthcare providers who were sometimes unable to answer to the large number of home visit requests, and reported difficulties in abiding by social distancing rules in homes. A midwife in the United Kingdom mentioned that "*All women were called prior to visits—quite time consuming. Basic PPE [personal protective equipment] for all visits—hot, time consuming putting on and taking off. Hard to social distance especially in small accommodation.*" Newborn routine consultations were also affected by the pandemic, as a paediatrician in Belgium described: "*it is noteworthy that up to now there remain delays in the follow-up for postnatal control, some infants have not seen a paediatrician before the age of 2 months*".

From the care availability side, shortening of operating hours, reducing number of days of care provision, or completely suspending provision were not frequently reported adaptations (Fig 3). Some respondents mentioned in the open-text responses that the provision of postnatal group consultations, such as group childcare support and emotional support, was suspended during the month preceding the survey.

**3- Changes in the content of postnatal care.** The adaptations made to the content of PNC are divided into changes to the care provided to all women and newborns (Fig 4), and changes relevant to women and newborns with suspected or confirmed COVID-19 (Fig 5).

Several elements of care in the routine PNC package were reportedly reduced or suspended in the month preceding the survey (Fig 4). About 34% (n = 14/41) of respondents from low-income countries reported that the provision of postnatal family planning counselling was reduced compared to 16% (n = 15/95) of respondents in high-income countries. On the other hand, reduction in the provision of breastfeeding support was reported by 40% (n = 38/95) of respondents in high-income countries, compared to 7% (n = 3/41) in low-income countries. Other reductions in care content include less social care support or referral (n = 64/266; 24%), mental health monitoring and support to women (n = 60/266; 23%), newborn weight monitoring (n = 56/266; 21%) and newborn vaccination (n = 50/266; 19%; Fig 4), and a midwife reported reduced newborn audiology screening in the open-text responses.

Earlier discharge after birth in a health facility was the most commonly reported adaptation across all the three themes identified in this study (n = 183/310; 59%) consistently across the three country income groups. In the open-text responses, a few healthcare providers elaborated that the reasons for early discharge were either a change in practice, or because these were requested by women because of strict visiting rules. A midwife from Germany mentioned that: "[*b*]*ecause of the general prohibition on visitors in clinics, women went home after 6 hours post childbirth, so that meant more home visits.*" In low-income countries, 45% (n = 21/47) of respondents reported a reduction in the frequency of routine postnatal monitoring in the health facility before discharge, compared to 11% (n = 13/120) of respondents in high-income countries (Fig 4).

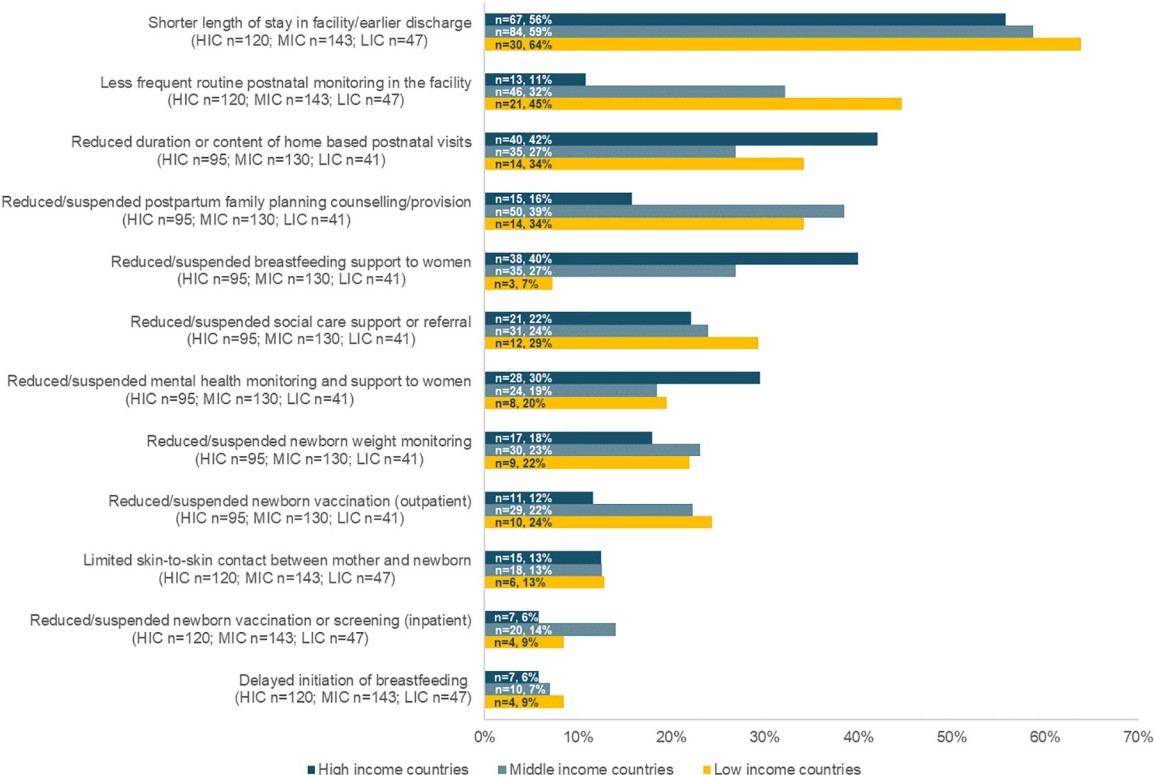

**Fig 4. Number and percentage of respondents reporting changes in postnatal care content, by country income level during the month preceding the survey.**

Although some respondents mentioned that they were providing more home visits (see theme 2 above), more than 30% (n = 89/266) of healthcare providers reported that the duration or the content of care provided during these home visits was reduced (Fig 4).

## Changes specific to women suspected/confirmed with COVID-19

The adaptations and restrictions made to breastfeeding support, skin-to-skin contact, and separation of mother-baby dyads were in many cases dependent upon the clients' COVID-19 status. For example, some respondents mentioned that skin-to-skin contact was encouraged among 'healthy' mothers, but was not allowed to be practiced with mothers with COVID-19. In other cases, the practice of skin-to-skin contact depended on availability of staff, as described an obstetrician/gynaecologist in Spain: "*In the case of caesarean sections, during the stay in the resuscitation unit, skin-to-skin could only be done if there were enough staff to deal exclusively with the woman and the newborn*". Almost one third (n = 36/120) of respondents from high-income countries noted that COVID-19 suspected/confirmed mothers were being separated from their newborns (Fig 5). Some facilities introduced a consent process and required women to sign a form in order to keep their newborns, waiving the hospital's liability in case of newborn's infection: "*Women are made to sign a 'consent' in the delivery room saying they agree to keep the baby exposing it to the possibility of infection transmission or leave it in a 'healthy' neo-isolate*", explained a midwife in Argentina. Even in facilities that have not yet received any COVID-19 maternity patients, guidance was put in place to separate mothers from newborns and not allow breastfeeding for COVID-19 confirmed mothers. A neonatologist from South Africa wrote: "*If mom [is] positive, and baby admitted to the neonatal unit,*

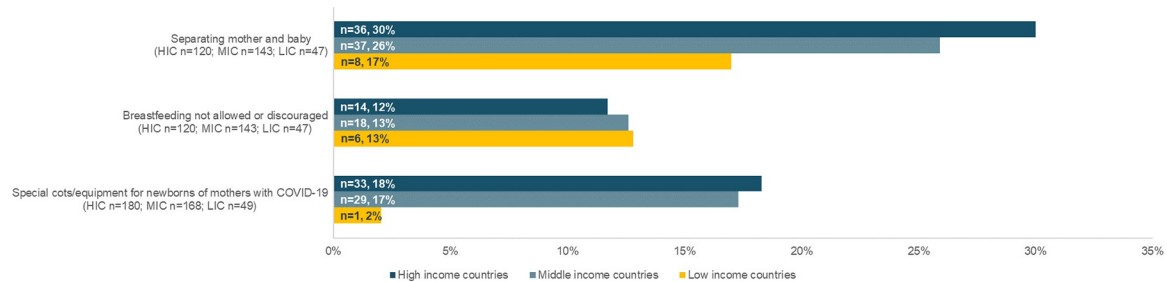

**Fig 5. Number and percentage of respondents reporting on postnatal care for newborns and mothers suspected or confirmed with COVID-19, by country income level during the month preceding the survey.**

*then no breastfeeding until 14 days isolation.*" In some cases, a negative PCR test was required before starting breastfeeding; a midwife in Argentina wrote: "*The possibilities of early start of breastfeeding are nil. . . Formula milk is administered to 90% of newborns, since the results of maternal swabs are not obtained in a timely manner, leaving mother and newborn separated during the first hours of puerperium.*" Only 2% (n = 1/49) of low-income country respondents mentioned that the facility where they worked dedicated special cots and equipment solely to be used for newborns of mothers with COVID-19, compared to 17% (n = 29/168) and 18% (n = 33/180) of respondents in middle- and high-income countries, respectively (Fig 5).

Fig 6 presents a synthesis of commonly reported changes and adaptations to PNC by country income group, and how they affected care availability, coverage, and quality during the COVID-19 pandemic.

## Discussion

This paper describes PNC service availability, content and quality during the COVID-19 pandemic, by triangulating quantitative and qualitative data from healthcare providers in

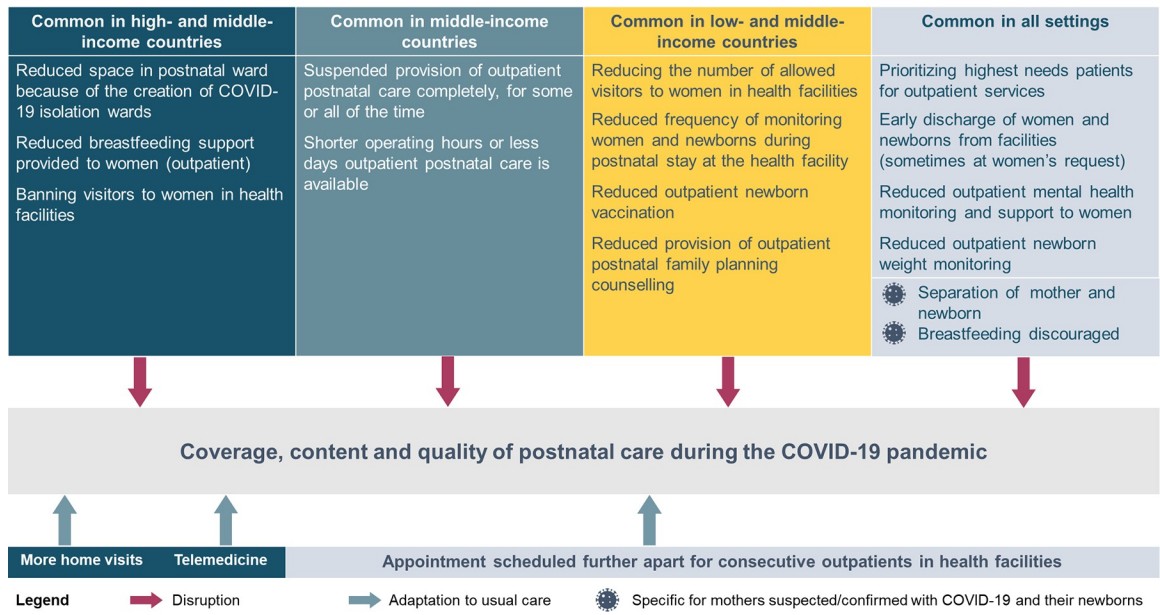

**Fig 6. Changes and adaptations to postnatal care provision and availability and their impact on care during the COVID-19 pandemic, by country income group.**

61 countries. Findings show significant disruptions to the content and quality of PNC, with notable differences by country income groups. The most common disruptions are shorter length-of-stay in health facilities after birth and banning birth companions and postnatal visitors to women and babies in health facilities. Availability and utilisation of PNC services were also affected as routine checks and preventive care to women and newborns reduced, and healthcare providers perceived a decline in the number of women and babies using PNC services. Our findings are in line with previous studies of healthcare providers conducted in the UK [41,42], Australia [43], and a global study focusing on newborn care services [17], that also show strict rules on companionship and postnatal visits to women and babies. Similarly, of 110 countries reporting to the second WHO pulse survey in the first quarter of 2021, 33% reported disruptions to PNC services for women and newborns, with 2% reporting more than a 50% disruption [22]. A survey of health facilities in the UK showed that 56% of healthcare sites reported a reduction in the number of postnatal contacts [42].

The disruptions documented in this analysis are concerning for three main reasons. First, coverage and quality of PNC were sub-optimal long before the pandemic, being among the worst performing indicators in the reproductive, maternal, newborn, and child health continuum of care [10,11,44]. Therefore, any further decrease in effective PNC coverage and quality could contribute to increased (preventable) morbidities among mothers and newborns due to insufficient early detection. Disruptions to PNC seem to be accumulating over the period of the pandemic, which is likely to extend for a long time. This adds a layer of protracted challenges that have to be overcome to achieve any future progress in this area. Second, disruptions to PNC services were more commonly reported by respondents in low-income countries where the burden of maternal mortality, morbidity and long-term health outcomes including maternal mental health are highest, and where maternal mental health issues are less detected and addressed due to limited mental health infrastructure [2,7,45]. Pre-existing socio-demographic, geographic, and financial barriers to access to PNC in low-income countries [46,47], were compounded by lockdowns, closure of health facilities, and reductions in frequency and content of care during the pandemic. Third, disruptions have continued to exist beyond the initial phase of the pandemic, and are not treated as short-term solutions as part of the emergency response. Clear recommendations and guidelines were available, yet suboptimal adaptations and processes continued to be in place. This is particularly worrying for low- and middle-income countries with slow COVID-19 vaccination programs [48], where re-implementation of lockdowns increases the likelihood of keeping these disrupting measures for a sustained period of time [49,50]. If interruptions of essential services, including PNC, continue to worsen throughout the pandemic, we are likely to lose previously achieved progress in protecting women's and newborns' lives and wellbeing [51].

Separation of parents and newborns (including for newborns who are in the NICU), absence of skin-to-skin contact, and reduced support for early and sustained breastfeeding were common practices for care provided to women suspected or diagnosed with COVID-19 and their babies. At the time of data collection, the evidence was clear that the benefits of breastfeeding and skin-to-skin outweigh the risk of infection with SARS-CoV-2, with several national and international guidelines published [38,52–54]. A review of guidelines from 33 countries on the care of infants whose mothers were suspected or confirmed with COVID-19 found that recommendations against practices supportive of breastfeeding were common [31]. Additionally, national guidelines were rapidly changing during the pandemic, which could lead to confusion, distrust, or mis-information among healthcare providers [14]. Concerted advocacy with community engagement is needed to enhance timely reach of international and national evidence-based guidance to healthcare providers. In that regard, concrete plans to

support the translation of knowledge into action from the national to the facility level and strengthening clear monitoring and reporting mechanisms are required to ensure that denial of essential recommended practices does not persist.

The WHO recommends a minimum hospital stay of 24 hours after uncomplicated vaginal birth to allow sufficient time to monitor the mother and the newborn, assess critical complications, and provide the needed counselling on childcare, detection of danger signs, breastfeeding, among others [1]. In this study, a reduction in postpartum length-of-stay was reported by 60% of respondents, yet the "new" duration was not captured. Considering the variability in national- and facility-level guidelines on length-of-stay, it is not possible to understand whether the "new" duration was below or above the recommended 24 hours. Before COVID-19, the average length-of-stay was already as low as half a day in some low- and middle-income countries [10]. In some resource-limited and congested settings, women and newborns get discharged or leave as quickly as 6 hours after birth due to lack of space and comfort in the facility [55]. We cannot with our data confirm whether these previously short durations were reduced during COVID-19, yet it is critical to note that any reduction in postnatal monitoring is a threat to the health of the mother and newborn, and could lead to preventable morbidities and mortalities, particularly in the case of complicated childbirth or birth by caesarean section. Early discharge after birth–during a pandemic or not–should be accompanied by close follow-up and monitoring at home [25,56,57].

Beyond length-of-stay, we identified serious compromises to PNC quality during this period of the pandemic. Previous research using the same data source documented gaps in provision of respectful care to women and their newborns along the maternity care continuum [14]. Denial of women's preferences to practice early breastfeeding and skin-to-skin contact and have their preferred companions during the immediate postpartum period: 1) are against the recommendations of respectful maternity care, and 2) erode women's and families trust in the health system, and 3) could have been substantially minimised by adhering to guidelines that existed before and that were developed during the COVID-19 pandemic [29,30,39]. Research shows that high quality of care during antenatal and intrapartum care, and the immediate postnatal period is associated with the utilisation of the long-term PNC services [58]. PNC coverage and health-seeking is therefore at risk of declining, during and beyond the pandemic, in light of the cumulative effect of poor experiences of care along the continuum, which could lead to distrust in the health system. Additionally, women who gave birth during the pandemic suffer from worse mental health conditions such as anxiety and depression [19,59–62], and this could be a result of the reduction in social and emotional support [23,63,64]. Restricting visits only to "partners" or "spouses" discriminates against women whose companions of choice are their mothers, other family members or friends, and implies that some women are spending a lonely journey in health facilities, from start to end. This violates the WHO recommendations of a woman-centred, respectful care experience for all women during the pandemic.

On the other hand, positive adaptations to ensure the continuation of PNC provision safely were reported. In few settings, restrictions on visitors were lifted during this period showing a progressive adaptation to the evolving pandemic situation. Spacing appointments between consecutive women in postnatal outpatient clinics was implemented to reduce the crowding in waiting areas. This could allow more time for healthcare providers to provide comprehensive, respectful and individualised care to the woman and her family. Some practices are also favoured by women; for example the absence of visitors allowed more privacy to breastfeeding women [43], and some women preferred to leave the health facility early. A recent meta-synthesis of qualitative studies shows that postpartum women struggle with shared rooms, lack of privacy in health facilities, and rules on visiting hours [65]. Women's voices and preferences of

PNC, should be taken into consideration when devising national recommendations and contextual guidance on modality and content of care provision. Self-care strategies such as the training women for newborn weight monitoring was mentioned in our study. Capacity strengthening to support self-care interventions for mother-baby dyad is needed to ensure the continuity of PNC services during health system shocks and beyond.

## Differences in adaptations by country income groups

The discrepancies in the reported adaptations by country income group should be interpreted with special attention to the context, resource availability, modalities of care provision that existed before the pandemic, and in light of the epidemiology of COVID-19 at the time of data collection. Differences related to COVID-19 infection prevention measures were more commonly reported in high-income countries compared to low-income countries, such as the creation of isolation wards leading to reduced bed/space availability, and the allocation of cots for newborns of COVID-19 confirmed mothers. These can be a result of a pre-existing shortage of resources with a relatively higher fertility rate in low-income countries, where many postnatal wards are extremely congested. Or this can be explained by the varying epidemiology of COVID-19 between the countries, whereby the spread was slower in some low-income countries at the time of data collection. In many low-income countries, including in sub-Saharan Africa, visitors to women in health facilities after birth mostly bring money, supplies and food [66]–commodities which are not offered by health facilities. This might have been taken in consideration when designing mitigation measures in low-income countries, where mostly visitors were reduced and not banned, to keep allowing women access to necessary supplies. Despite its limitations, such as the increased probability of overlooking postpartum danger signs in the absence of physical examination [23], telemedicine use was recommended to ensure the safe continuation of care provision during the pandemic [29,56,57]. Additionally, women reported encountering technology-related challenges when using of virtual PNC in Canada [24]. Telemedicine was not commonly reported in low- and middle-income settings, for PNC as for other maternal health services [15], and is a clear expression of the ongoing digital divide and global inequality in access to information and communication technology [67,68].

Home-visits were recommended by WHO to ensure continuity of PNC during lockdowns, and increase routine monitoring in cases of early discharge [29,56,57]. Certain settings benefit from good community health systems with trained community health workers and support systems that can deliver quality maternal health services as part of home visitation programmes. In our survey, home visits were limited to high-income countries where they were common before the pandemic, such as Belgium, Germany, Canada, and the UK [69–72]. These health systems capitalized on existing infrastructure to de-congest healthcare facilities and reduce risks of nosocomial infections while ensuring care continuity. Healthcare workers who provide care at home, mostly midwives or trained community health workers, faced many challenges since they were not considered frontline workers, and had inadequate access to personal protective equipment [16,73]. These barriers could have been exacerbated for healthcare providers in low-income countries if home visits were to be "newly" introduced during the uncertain conditions of a pandemic. Overarching global recommendations for mitigation strategies must be contextualised vis-à-vis the availability of resources and infrastructure in each setting. Innovative and context-specific alternatives must be developed by including voices from these healthcare systems in the decision making in order to prevent that essential services, such as PNC, fall through the cracks, and to avoid worsening the inequalities in access to preventive and life-saving care [74].

## Limitations

The sampling strategy for the online survey and the non-existence of a sampling frame means that the data are not representative nor generalizable; we rather intend to describe the situation in-depth as reported by respondents. For this sub-analysis on PNC, the sample size was limited to healthcare providers who answered the optional module, i.e. might be biased and underrepresent providers who may have been pre-occupied with less time to answer more questions; however, the characteristics of the sub-sample in terms of country, cadre, job, are similar to those of the full sample. About one quarter of the respondents who answered the PNC questions did not provide services in the postnatal period. S3 Table presents the findings disaggregated by whether respondents provided PNC or not, showing that for the majority of the outcomes, the results are comparable between the two groups. This indicates that maternity healthcare workers who are not directly involved in PNC report similar answers as those who are. This can be explained by the fact that maternal healthcare providers have shared experiences and are uniformly knowledgeable about care processes through day-to-day interactions between colleagues. The way in which the survey was designed limited the opportunity to have additional useful details, particularly on the new length-of-stay in the hospital after birth, or about the exact context in which the respondents work, such as previous length-of-stay before the pandemic, previous practices related to birth companions and visitors, etc. Future research should include questions that explore these areas in more detail. The use of the World Bank classification of countries' income groups also has its limitations because it ignores health system related factors that could influence the response to the pandemic. Also, the sampling strategy was not stratified by country-income group and therefore the data are not representative of the three groups included in this analysis, and that is why the comparisons were limited to descriptive statistics and no further statistical tests were conducted.

## Conclusion

COVID-19 not only affects the health of those who are infected with the virus, but is also a threat to all women and newborns, as both immediate and long-term PNC continued to be disrupted beyond the early phase of the pandemic. Before the pandemic, PNC coverage was among the lowest across the continuum of maternal and newborn healthcare, and issues with PNC quality were dominant. Issues such as maternal mental health, postnatal length-of-stay, visitors and companions, and non-separation were negatively affected without support by evidence. Advocacy is required to prioritise the protection of these aspects of care during events that disrupt the regular functioning of health systems. At a time beyond the first wave of the pandemic, access to the adequate knowledge and time to provide an evidence-based response should have been available to decision-makers in order to maintain the provision of essential services. It is important to document and disseminate lessons and experiences from the settings where initial disruptions were short-lived and where care was rapidly adapted to most recent evidence. Service integration could be a potential solution to ensuring the provision of the full PNC package with one contact, while reducing the risk of COVID-19 transmission. These lessons must turn into advocacy in order to raise the bar for PNC policy and service delivery especially in low- and middle-income countries, during and beyond the COVID-19 pandemic, and prioritise emergency preparedness to maintain this essential service.

## Supporting information

**S1 Table. Survey questions included in the analysis.**
(DOCX)

**S2 Table. Country distribution of maternal and newborn healthcare providers (n = 424).**
(DOCX)

**S3 Table. Analysis disaggregated by type of care provided by respondents (PNC vs no PNC).**
(DOCX)

## Acknowledgments

We would like to thank the maternal and newborn healthcare providers who contributed their valuable time to respond to the survey during the second round, despite ongoing difficult circumstances and high workload. We thank all study collaborators and colleagues who supported in questionnaire development, translation and played a key role in distributing the invitation for this survey. We also acknowledge the Institutional Review Committee at the Institute of Tropical Medicine for providing helpful suggestions on this study protocol, and for the expedited review of this study.

## Author Contributions

**Conceptualization:** Aline Semaan, Teesta Dey, Amani Kikula, Anteneh Asefa, Thérèse Delvaux, Etienne V. Langlois, Thomas van den Akker, Lenka Benova.

**Data curation:** Aline Semaan.

**Formal analysis:** Aline Semaan.

**Funding acquisition:** Lenka Benova.

**Methodology:** Aline Semaan.

**Visualization:** Aline Semaan.

**Writing – original draft:** Aline Semaan.

**Writing – review & editing:** Aline Semaan, Teesta Dey, Amani Kikula, Anteneh Asefa, Thérèse Delvaux, Etienne V. Langlois, Thomas van den Akker, Lenka Benova.

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
