## [Decision Letter · Decision Letter 0]

13 Oct 2021

PGPH-D-21-00630

“Separated during the first hours” - Postnatal care for women and newborns during the COVID-19 pandemic: A mixed-methods cross-sectional study from a global online survey of maternal and newborn healthcare providers

Dear Dr. Semaan,

Thank you for submitting your manuscript to PLOS Global Public Health. After careful consideration, we feel that it has merit but does not fully meet PLOS Global Public Health’s publication criteria as it currently stands. Therefore, we invite you to submit a revised version of the manuscript that addresses the points raised during the review process.

We look forward to receiving your revised manuscript.

Kind regards,

Jenil Patel, MBBS, MPH, PhD

Academic Editor

Journal Requirements:

1. Please provide separate figure files in .tif or .eps format only, and remove any figures embedded in your manuscript file.  If you are using LaTeX, you do not need to remove embedded figures.

2. We have noticed that you have uploaded supporting information but you have not included a list of legends.  Please add a full list of legends for all supporting information files (including figures, table and data files) after the references list. 

Reviewers' comments:

Reviewer's Responses to Questions

**Comments to the Author**

1. Does this manuscript meet PLOS Global Public Health’s publication criteria? Is the manuscript technically sound, and do the data support the conclusions? The manuscript must describe methodologically and ethically rigorous research with conclusions that are appropriately drawn based on the data presented.

Reviewer #1: Yes

Reviewer #2: Yes

Reviewer #5: Yes

2. Has the statistical analysis been performed appropriately and rigorously?

Reviewer #1: I don't know

Reviewer #2: Yes

Reviewer #5: N/A

3. Have the authors made all data underlying the findings in their manuscript fully available (please refer to the Data Availability Statement at the start of the manuscript PDF file)?

Reviewer #1: Yes

Reviewer #2: Yes

Reviewer #5: No

4. Is the manuscript presented in an intelligible fashion and written in standard English?

Reviewer #1: Yes

Reviewer #2: Yes

Reviewer #5: Yes

5. Review Comments to the Author

Reviewer #1: Dear author,

Thank you for the opportunity of reviewing your manuscript.

Even though the subject is of interest to me, I did not quite understand what you tried to show in this article. Indeed initial PNC coverage i.e. before the pandemic was not studied it is therefore complicated to conclude that COVID 19 has modified this follow-up. Moreover, the consequences of the potential modification in PNC have not been studied.

Specific comments follow;

Lines 111-112 "We present findings from a global online survey conducted after the first wave of the COVID-19 pandemic and the concomitant emergency response." This sentence is unclear, please clarify what the authors mean by concomitant emergency response.

The introduction is too long, gives both a lot of general data with no link to the study (lines 56-73) and very (too?) specific data (lines 85-95). At the end of the introduction the purpose of the study and what it will add to literature is imprecise.

The objective of the study is very vague, the aim was just to describe how PNC was organized during the COVID 19 pandemics? If so, why compare countries according to their incomes?

Method section: the lack of details on how the analyses were conducted did not help me understand the objective of this study.

I am quite puzzled after reading this article, the data is interesting but the way this article is written does not highlight the usefulness of these data.

Reviewer #2: This paper describes postnatal care service availability, content and quality during the COVID-19 pandemic, using quantitative and qualitative data from healthcare providers in 61 countries. Results show significant disruptions to the content and quality of postnatal care, with notable differences by country income groups. These are important findings and the manuscript is well written. The analysis is descriptive, which is adequate. There are only few inaccuracies to consider:

Page 6, line 159: 443 out of 1405 is not 30% (but 31.5%)

Page 6, line 160: 424 out of 443 is not 99% (but 95.7%)

Page 10, line 205: Please provide absolute numbers (n) for these percentages.

Page 11, Box 1: 91 out of 346 is not 35% (but 26.3%)

Page 11, line 230: Text says 36%, figure says 37%. Please correct.

Page 12, lines 243 + 250: Please provide absolute numbers (n) for these percentages.

Page 13, lines 282 + 283 + 294: Please provide absolute numbers (n) for these percentages.

Page 13, line 290: Text says 44%, figure says 45%. Please correct.

Page 14, line 316: "the facility were they worked" should be "the facility where they worked". Please correct.

Figures 1 to 4: Please provide absolute numbers (n) for all percentages in these figures.

Please revise and check all remaining numbers and percentages twice!

Reviewer #5: This study used a questionnaire with a large number of healthcare providers (424) across 61 countries to determine if there were changes in routine post-natal care due to Covid-19. The abstract is clear and concise. They claim and provide evidence for a reduction in postnatal care services and provide a good set of evidence that these reductions could increase the risks of poor health outcomes. The data supports the claims generally apart from some minor issues mentioned below. The questions used in the survey are provided which is a nice change and enough information is given about the methods used for reproducibility.

Original data was not provided due to some small cell counts. I am not sure that the anonymity of respondents would be truly affected unless there was less than 5 workers at a facility. The authors might consider releasing data from larger institutions although I am not sure if that would ultimately provide much as the results section is comprehensive. How this study fits into the existing literature is dealt with in depth in both the Introduction and Discussion.

The manuscript is well organised and understandable. The Tables and Figures are useful.

A few specific thoughts which might help explain a few details for the reader are listed below.

Abstract: Mentions “Early initiation of breastfeeding was delayed due to waiting for maternal SARS-CoV-2 test results.” Figure 3 suggests that this affected <10% iof countries. However, in the text at line 280 it says “On the other hand, reduction in the provision of breastfeeding support was reported by281 40% of respondents in high-income countries, compared to 7% in low-income countries”. For me this second set of facts would be more relevant for inclusion in the abstract as it was reported by a larger proportion of respondents.

Line 85: Looking at Table 9 in the reference provided [22] it appears to show that 33% of countries reported disruption in post-natal care with 2% more than 50% disrupted. I would encourage the authors to check the accuracy of the sentence provided. This fact is again repeated at line 339 in the discussion which may also need to be altered.

Line 160: You state that you included data from 424 of 443 providers that responded to the optional questionnaire component. To me that comes out to be about 95.7% rather than the 99% reported. Is the n or the proportion reported incorrectly?

Line 175: How were final decisions made about what codes to use? Consensus? Majority?

Line 188: If a quarter of respondents didn’t provide any PNC how were they qualified to comment on the PNC provided? Was it because they were managers or heads of team or something similar? It may increase confidence in the readers if you could make this clearer. Have now seen that this is addressed at line 470 that they may have had contact with other staff who could have shared information with them. I feel that hearsay evidence would be less reliable than expected from a questionnaire of this nature. Would it be possible to provide any further analysis/evidence of the similarity of data provided by the cohort which wasn’t involved with PNC directly compared with those that were to ensure consistency?

Line 330: “Findings show significant disruptions to the content and quality of PNC, with notable differences by country income groups.” This is similar to what is mentioned in the abstract. However, at line 242 states “Changes to PNC availability and use were not commonly reported and mainly included the prioritisation of providing care to patients with the highest needs (33%)” .Perhaps you could make it clearer in the abstract that content and quality was affected but not nearly as much as availability and use.

Line 344 it states: “Therefore, any further decrease in effective PNC coverage and quality could contribute to increased undetected preventable morbidities among mothers and newborns, and risk losing the ability of achieving future progress in this area.” It is not clear to me why an increase in morbidities would be undetectable. Could you make this clearer. Secondly, what mechanism would cause a risk of losing the ability to achieve future progress once things get back to ‘normal’? Again, a bit more explanation might make the proposed links clearer.

Line 388 and accompanying paragraph. Some information seems repeated here (e.g. consequences of reduction in contact with companions mentioned across five/six sentences.) If there is any need to reduce word-count it might be achieved by combining/condensing this section.

Line 471: Here limitations are being discussed. “The nature of the survey and using close-ended questions limited……length of stay questions. It seems like the survey would be something that should be partly under the control of the researchers and could be captured in close-ended questions anyway. Perhaps it would be better just to acknowledge that length of stay questions would be good to incorporate in the future?

I enjoyed reading this paper and I think it makes a good contribution. Thanks for your hard work.

6. PLOS authors have the option to publish the peer review history of their article (what does this mean?). If published, this will include your full peer review and any attached files.

**Do you want your identity to be public for this peer review?** For information about this choice, including consent withdrawal, please see our Privacy Policy.

Reviewer #1: No

Reviewer #2: No

Reviewer #3: No

Reviewer #4: No

Reviewer #5: No

---

## [Decision Letter · Decision Letter 1]

19 Jan 2022

PGPH-D-21-00630R1

“Separated during the first hours” - Postnatal care for women and newborns during the COVID-19 pandemic: A mixed-methods cross-sectional study from a global online survey of maternal and newborn healthcare providers

Dear Dr. Semaan,

Thank you for submitting your manuscript to PLOS Global Public Health. After careful consideration, we feel that it has merit but does not fully meet PLOS Global Public Health’s publication criteria as it currently stands. Therefore, we invite you to submit a revised version of the manuscript that addresses the points raised during the review process.

Please address the two minor comments below:

Please ensure that your decision is justified on PLOS Global Public Health’s publication criteria and not, for example, on novelty or perceived impact.

We look forward to receiving your revised manuscript.

Kind regards,

Jenil Patel, MBBS, MPH, PhD

Academic Editor

Journal Requirements:

1. Please update your Competing Interests statement. If you have no competing interests to declare, please state: “The authors have declared that no competing interests exist.”

Additional Editor Comments (if provided):

Minor revision:

Please address the following two comments:

1. Many thanks for the revised version of the manuscript, which addresses all of the reviewers' comments. I am pleased that you have included absolute numbers and checked the percentages. There is only one minor point to consider left. Table 1 now includes the numbers of data available for each section. Please also update the numbers in the header of the table (n=193 etc). You may further include this information for each section of the table. Many thanks!

2. It is a study conducted painstakingly. The authors have answered all the queries put forward by the reviewers in a detailed manner. However the study is quite exhaustive and after reading this article, i am confused as to what message are the authors trying to put forth. It would be better if the conclusion is more precise as to throw points which will help in clinical practice.

Reviewers' comments:

Reviewer's Responses to Questions

**Comments to the Author**

1. If the authors have adequately addressed your comments raised in a previous round of review and you feel that this manuscript is now acceptable for publication, you may indicate that here to bypass the “Comments to the Author” section, enter your conflict of interest statement in the “Confidential to Editor” section, and submit your "Accept" recommendation.

Reviewer #2: All comments have been addressed

Reviewer #3: All comments have been addressed

2. Does this manuscript meet PLOS Global Public Health’s publication criteria? Is the manuscript technically sound, and do the data support the conclusions? The manuscript must describe methodologically and ethically rigorous research with conclusions that are appropriately drawn based on the data presented.

Reviewer #2: Yes

Reviewer #3: Yes

3. Has the statistical analysis been performed appropriately and rigorously?

Reviewer #2: N/A

Reviewer #3: I don't know

4. Have the authors made all data underlying the findings in their manuscript fully available (please refer to the Data Availability Statement at the start of the manuscript PDF file)?

Reviewer #2: No

Reviewer #3: Yes

5. Is the manuscript presented in an intelligible fashion and written in standard English?

Reviewer #2: Yes

Reviewer #3: Yes

6. Review Comments to the Author

Reviewer #2: Many thanks for the revised version of the manuscript, which addresses all of the reviewers' comments. I am pleased that you have included absolute numbers and checked the percentages. There is only one minor point to consider left. Table 1 now includes the numbers of data available for each section. Please also update the numbers in the header of the table (n=193 etc). You may further include this information for each section of the table. Many thanks!

Reviewer #3: It is a study conducted painstakingly. The authors have answered all the queries put forward by the reviewers in a detailed manner. However the study is quite exhaustive and after reading this article, i am confused as to what message are the authors trying to put forth. It would be better if the conclusion is more precise as to throw points which will help in clinical practice.

7. PLOS authors have the option to publish the peer review history of their article (what does this mean?). If published, this will include your full peer review and any attached files.

**Do you want your identity to be public for this peer review?** For information about this choice, including consent withdrawal, please see our Privacy Policy.

Reviewer #2: No

Reviewer #3: No

---

## [Editor Report · Decision Letter 2]

23 Feb 2022

“Separated during the first hours” - Postnatal care for women and newborns during the COVID-19 pandemic: A mixed-methods cross-sectional study from a global online survey of maternal and newborn healthcare providers

PGPH-D-21-00630R2

Dear Ms Semaan,

We are pleased to inform you that your manuscript '“Separated during the first hours” - Postnatal care for women and newborns during the COVID-19 pandemic: A mixed-methods cross-sectional study from a global online survey of maternal and newborn healthcare providers' has been provisionally accepted for publication in PLOS Global Public Health.

Best regards,

Jenil Patel, MBBS, MPH, PhD

Academic Editor